# Quantitative Three-Dimensional Computed Tomography Measurements Provide a Precise Diagnosis of Fractures of the Mandibular Condylar Process

**DOI:** 10.3390/jpm12081225

**Published:** 2022-07-27

**Authors:** Enkh-Orchlon Batbayar, Nick Assink, Joep Kraeima, Anne M. L. Meesters, Ruud R. M. Bos, Arjan Vissink, Max J. H. Witjes, Baucke van Minnen

**Affiliations:** 1Department of Oral and Maxillofacial Surgery, School of Dentistry, Mongolian National University of Medical Sciences, Zorig Street, Ulaanbaatar 14210, Mongolia; 2Department of Surgery, University Medical Center Groningen, University of Groningen, 9713 GZ Groningen, The Netherlands; n.assink@umcg.nl (N.A.); a.m.l.meesters@umcg.nl (A.M.L.M.); 33D Lab, University Medical Center Groningen, University of Groningen, 9713 GZ Groningen, The Netherlands; j.kraeima@umcg.nl (J.K.); m.j.h.witjes@umcg.nl (M.J.H.W.); 4Department of Oral and Maxillofacial Surgery, University Medical Center Groningen, University of Groningen, 9713 GZ Groningen, The Netherlands; r.r.m.bos@umcg.nl (R.R.M.B.); a.vissink@umcg.nl (A.V.)

**Keywords:** mandibular fractures, mandibular condyle, three-dimensional imaging, classification, diagnoses

## Abstract

As 2D quantitative measurements are often insufficient, a standardized 3D quantitative measurement method was developed to analyze mandibular condylar fractures, and correlate the results with the mandibular condylar fracture classifications of Loukota and Spiessl and Schroll and clinical parameters. Thirty-two patients with a unilateral mandibular condylar fracture were evaluated using OPT, 2D (CB)CT images, and 3D imaging to measure the extent of the fractures. The maximum mouth opening (MMO) was measured. Ramus height loss could be measured only in OPT, but not in 2D CT images. The Intraclass Correlation Coefficient was excellent in the 3D measurements. In the Loukota classification, condylar neck fractures had the largest median 3D displacement and the highest rotations of the fracture fragments. The largest fracture volume was observed in base fractures. According to the Spiessl and Schroll classification, type V fractures had the largest median 3D displacement and the highest rotation in the *X*-axis and *Z*-axis. Type I fractures had the largest fracture volume. We found a moderate negative correlation between MMO and 3D displacement and rotation on *Z*-axis. The 2D quantitative analysis of condylar fractures is limited, imprecise, and not reproducible, while quantitative 3D measurements provide extensive, precise, objective, and reproducible information.

## 1. Introduction

Fractures of the condylar process represent 29–52% of all mandibular fractures. However, there is a lack of consensus on treatment of these fractures [1,2,3,4,5]. Generally speaking, practitioners chose one of two treatment modalities for condylar fractures: surgical (open) treatment or conservative (closed) treatment [4,6]. The choice of treatment modality can be based on the classification and characteristics of the fracture [4], but in practice, this choice is usually based on the training, experience, and skills of the surgeon. To overcome these differences in approach, several classification systems for condylar fractures have been proposed [7,8,9,10], of which the classifications of Loukota [11,12] and Spiessl and Schroll [13] are the most widely used.

To diagnose and classify condylar fractures, practitioners frequently use conventional two-dimensional (2D) imaging, such as panoramic radiographs (OPT) and Towne projections. Based on these radiographs, fracture characteristics such as ramus height loss and deviation of the condylar head are measured to decide on the preferred treatment modalities [14,15,16,17]. However, Kommers et al. [18] questioned the reliability of measuring the ramus height loss of fractures of the condylar process on OPT. Additionally, others showed that measurements on conventional 2D imaging are highly observer dependent and vary with patient positioning when taking X-rays [19,20,21,22,23].

Despite the efforts of several authors, no consensus has been reached on clinically relevant universal classification and subsequent treatment choices of condylar fractures. This situation continued even after the introduction of computed tomography (CT) [7,9,24]. Such a classification is needed because the description of condylar fractures is currently dependent on 2D slices of CT images, while the availability of conventional 2D images such as OPT and Towne is rapidly declining due to the introduction of low dose CT and the conebeam CT (CBCT) [7]. Even though CT and CBCT can display the anatomy in 3D, most CT data are still viewed in 2D slices. A major drawback of the use of 2D CT images is that the assessment of condylar fractures with this type of radiograph is also dependent on the skills of the clinician who analyses these 2D CT images and can vary depending on the CT characteristics such as resolution, slice thickness, and field of view [7]. Moreover, measurements such as the estimated loss of ramus height and deviation of the condylar head are not reliable if they are based on 2D CT images [7].

Nowadays, quantitative three-dimensional (3D) CT measurements are used for diagnosis and classification of fractures of the tibial plateau, acetabulum, and radial heads [25,26,27,28]. These measurements have been proven reliable and have provided insight into the multidirectional aspects of these fractures based on this detailed information about fracture extent [25,26,27]. Therefore, the aim of this study was to develop a standardized 3D quantitative measurement method for mandibular condyle fractures with high inter-user reliability, to correlate the obtained results with the Loukota [11,12] and Spiessl and Schroll [13] classifications and with clinical parameters as ramus height loss and maximum mouth opening.

## 2. Materials and Methods

### 2.1. Study Population

All consecutive patients that were diagnosed with a unilateral mandibular condylar fracture with or without concomitant other mandibular fractures at University Medical Center Groningen between 2015 and 2018 were reviewed. Patients whose diagnosis was confirmed with CBCT or CT with a maximum slice thickness of 1 mm were included. Exclusion criteria were a bilateral condylar fracture and unavailability of a diagnostic CT or CBCT scan. This study was conducted according to the guidelines of the Declaration of Helsinki. There is no file number of the medical ethical committee as it is a retrospective study that analyses anonymous CT data from our center only. The default is that this does not require a review board statement.

### 2.2. Fracture Classification and Clinical Parameters

To classify the condylar fractures, based on the preoperative standard 2D format CT or CBCT scans in axial, coronal, and sagittal reconstructions, the classifications according to Loukota [11,12] and Spiessl and Schroll [13] were used.

Due to the retrospective nature of this study, not all 32 eligible cases were completely documented with respect to clinical parameters. maximum mouth opening (MMO) measurements (from incisor to incisor) were available from 20 patients. These measurements consisted of MMO scores in mm on the same day (±1 day) that the CT or CBCT scan was taken.

In patients whose OPT and/or Towne projections were taken at the same time as the CT scan (±1 day), 2D measurements were made of ramus height loss and deviation of the fracture. Ramus height loss (2D measurement) was measured in millimeters, and deviation was measured in degrees according to the method described by Palmieri et al. [17].

### 2.3. Segmentation and 3D Model

Of all included patients, the CT or CBCT files were imported into the Mimics Medical software package (Version 21.0, Materialise, Leuven, Belgium) to create a 3D bone model of the skull and each part of the fractured mandible. The segmentation process was performed using a default bone threshold (Hounsfield Unit ≥ 226) followed by manual optimization by a trained observer (E.-O.B). All the segmented models were given a different color (Figure 1).

Next, the condylar fracture was virtually reduced to its anatomical position in 3-matic Medical (version 13.0; Materialise, Leuven, Belgium). As a guide for reduction, the contralateral non-fractured side was mirrored and aligned with the affected side (Figure 2).

### 2.4. The 3D Measurements

To measure the condylar fracture extent, the following new 3D measurements were used: 3D displacement, the volume of the fracture fragment and the rotation of the fracture fragment.

The 3D displacement. The 3D displacement of the fracture is the difference between the fractured and the reduced position of the condyle. This was assessed using 3-matic Medical and Matlab (R2014B, Mathworks, Natick, MA, USA) software. The extent of displacement/dislocation of the fragment(s) of the condyle was determined by calculating the 3D displacement along the *X*, *Y*, and *Z* axes. To calculate the 3D displacement, the fracture was virtually reduced in the 3D model, as described above (Figure 2). For each point on the surface of the 3D model, a difference in Euclidian distance between the fractured and reduced position was calculated in millimeters (mm). This 3D displacement of each part of the fragment is presented as a distance map in Figure 3, and was calculated as follows:$$3D\;\mathrm{displacement}=\frac{\sum_{i=1}^{n}\left(Surface_{i}*\;displacement_{i}\right)}{\sum_{i}^{n}\left(Surface_{i}\right)+Nondisplaced\;surface}$$

### 2.5. Volume of Fracture Fragment

The volume of the fracture fragment is the actual size of the fractured part of the condylar process. It was calculated in cubic millimeters (mm^3^) using 3-matic Medical.

The 3D rotations. The 3D rotation is a rotation of the fractured fragment of condyle in the *X*, *Y*, and *Z* axes. This was assessed using the same software (3-matic Medical and Matlab). This measurement was calculated in degrees (°). Because the orientation of the mandible within the original CT scan is greatly influenced by the positioning of the patient in the scanner, a standardized axis was defined using three landmarks on the bilateral lingulae of the mandible (Figure 4). This method has good reproducibility [29].

Deviation in axial alignment was described as the rotation difference along the *Y*-axis between the fragment and its original position, whereas the rotation difference along the *Y*-axis described the deviation in coronal alignment (Figure 5).

### 2.6. Reproducibility and Statistical Analysis

The condylar fractures were classified by two independent observers (E.-O.B and R.R.M.B.). Consensus was reached after discussion, without need of a third observer. Ramus bone height loss (2D) was measured by the same observers on the OPT.

3D displacement and 3D rotation were measured by one observer first (E.-O.B) in all 32 included patients. To assess the reproducibility, the 3D measurements were then performed by another independent observer (N.A.) in 10 patients, selected by E.-O.B, so that all types of condylar fractures were represented.

All statistical analyses were performed using IBM SPSS Statistics for Windows, Version 23.0 (Armonk, NY, USA: IBM Corp.) software. Reproducibility was calculated using the Intraclass Correlation Coefficient (ICC), with a two-way random, single-measurement model with the absolute agreement.

Normality of the data distribution was tested with the Kolmogorov–Smirnov test, and median and interquartile range (IQR) were reported for not normally distributed variables, and means and standard deviation (SD) for normally distributed variables. The Kruskal–Wallis test was used to compare the results of the measurements (MMO, ramus height loss, and 3D) with the fracture classifications (Loukota and Spiessl and Schroll). The correlation between MMO and measurements (ramus height loss and 3D) was determined using Spearman’s correlation test. Statistical significance was set at *p* < 0.05.

## 3. Results

### 3.1. Descriptives

In total, 32 patients with unilateral condylar fractures met the inclusion criteria. Median age was 29 (21;50) years, and 23 (72%) patients were male. In total 9 patients (29%) had condyle fractures concomitant with other mandibular fractures: parasymphyseal (n = 7), body (n = 1), and angle (n = 1). The median number of fracture fragments of the condyle was 1 (1;3). The mean MMO was 22.0 ± 7.6 mm. A Towne projection was not available for any of the 32 included patients, so measurement of deviation was not possible; OPT were available for 8 patients.

According to the Loukota classification of condylar fractures, 9 patients had diacapitular fractures (type A, n = 4; type B, n = 3, type C, n = 2), 5 patients had neck fractures, and 18 patients had base fractures. According to the Spiessl and Schroll classification, 9 patients had non-displaced fractures of the condyle (type I, n = 9), 8 patients had low condylar fractures with displacement (type II, n = 8), 6 patients had low condylar fractures with dislocation (type IV, n = 6), 3 patients had high condylar fractures with dislocation (type V, n = 3), and 6 patients had intracapsular or diacapitular fractures (type VI A, n = 4; VI B, n = 2). The type III fracture was not observed in this study.

The median (IQR) of the overall 3D displacement of the condylar fractures was 5.3 mm (2.3;8.6). Median of the rotations for the fractures was: *X*-axis, 5.4° (−1.3;59.1); *Y*-axis, −0.8°(−9.2;4.5); *Z*-axis, 4.8° (1.1;12.0).

Eight of 32 patients had a pre-operative OPT, and ramus height loss was measured in these patients only. Median loss of height was 5.7 mm (1.6;8.2).

### 3.2. Reproducibility

The Intraclass Correlation Coefficient (ICC) for the validation of the Loukota classification was 0.98 (95% CI, 0.96;0.99), and of the Spiessl and Schroll classification was 0.99 (95% CI, 0.93;0.98).

The ICC for reproducibility of the ramus height loss (2D) measurement was 0.86 (95% CI, 0.45;0.97) between the two observers.

The ICC for 3D displacement measurement was 0.98 (95% CI, 0.86;0.99), and the fracture rotation measurements were *X*-axis 0.99 (95% CI, 0.96;0.98), *Y*-axis 0.96 (95% CI, 0.84;0.99), and *Z*-axis 0.98 (95% CI, 0.95;0.99).

### 3.3. Clinical Parameters by Diacapitular Fractures (Loukota Classification)

In total nine patients had diacapitular fractures of the mandibular condyle: four patients had type A fractures, three patients had type B fractures and two patients had type C fractures. Because no OPT was available, ramus height loss was could not be measured in any of the diacapitular fractures. MMO was scored higher in type A (n = 4, 24.5 mm) and type B (n = 2, 26.5 mm) fractures and lowest in type C fractures (n = 1, 9.0 mm).

### 3.4. The 3D Measurements by Diacapitular Fractures (Loukota Classification)

The larger 3D displacement was measured in (n = 2, 10.3 mm) type C fractures followed by type A (n = 4, 8.3 mm) and type B (n = 2, 1.3 mm) fractures. The type C diacapitular fractures had greater rotation on the *X* and *Z* axis (n = 2, 25.8°, and 13.8°) compared to type A (n = 4, 10.2°, and 7.5°) and type B (n = 2, 1.3°, and 4.1°) fractures. The mean volumes of the fracture fragments were 693 mm for type A fractures, 851 mm^3^ for type B fractures and 761 mm^3^ for type C fractures.

### 3.5. Clinical Parameters by Condylar Process Fractures (Loukota Classification)

Ramus height loss was measured only in the condylar base fractures (n = 8); and the median was 5.7 mm (16;8.2). Therefore, no comparison between the classes was performed.

MMO was documented in 21 patients with condyle fractures, consisting of diacapitular fractures (n = 7), neck fractures (n = 4), and base fractures (n = 10). Higher mean MMO was observed in the base fractures (24.1 mm) and diacapitular (22.2 mm) fractures, and lower mean MMO in the neck fractures (16.5 mm) fractures (Table 1).

### 3.6. The 3D Measurements of Condylar Process Fractures (Loukota Classification)

Condylar neck fractures had the largest median 3D displacement (n = 5, 10.7 mm) compared to the diacapitular (n = 9, 6.1 mm) and base fractures (n = 18, 5.0 mm) of the condyle, but a statistically significant difference was not observed between the classes.

Condylar neck fractures had the highest median rotation on all three axes; *X*-axis 6.3°, *Y*-axis −3.6° and *Z*-axis 19.7°. The rotation of the fracture fragments did not differ between the Loukota classes. The mean volume of the fracture fragment was as follows: diacapitular 761 mm^3^, neck 2140 mm^3^, and base 2226 mm^3^. A significantly larger volume of the fragment was observed in base fractures compared to diacapitular and neck fractures.

### 3.7. Clinical Parameters According to the Spiessl and Schroll Classification

The ramus height loss was measured in following fracture types only: type I, II, and IV. High loss of ramus was measured in type IV fractures (n = 4, 8.1 mm) and type II fracture (n = 1, 6.2 mm) whereas low in the type I fractures (n = 3, 1.0 mm).

MMO was measured in 21 patients. Higher MMO was measured in type II (n = 3, 25 mm) and type IV (n = 5, 25 mm) fractures, followed by type VI A (n = 4, 24.5 mm), type I (n = 6, 23 mm), type VI B (n = 1, 18 mm), and type V (n = 2, 12 mm).

### 3.8. The 3D Measurements According to the Spiessl and Schroll Classification

The largest median of 3D displacement was observed in type V (n = 3, 11.9 mm) fractures followed by type VI A (n = 4, 8.3 mm), type IV (n = 6, 6.8 mm), type II (n = 8, 5.4 mm), type VI B (n = 2, 3.5 mm), and type I (n = 9, 2.3 mm). The 3D displacement differed significantly in between the fracture types.

The highest rotation on *X* and *Z*-axis was observed in the type V fractures (31.8°, 20.8°). Type IV fractures had the highest rotation on *Y*-axis (−13°). Fracture rotation varied in the other types of condylar fractures (Table 2). Significantly higher fracture rotations were observed in the type V fractures (*X* and *Y*-axis).

The largest mean volume of the fracture fragment was observed in type I (2291 mm^3^) and the smallest volume in VI A (693 mm^3^) fractures. A significant difference was observed between the fracture types.

### 3.9. Correlations between Clinical Parameters and 3D Measurements

The correlations between MMO and ramus height loss and 3D measurements are shown in Table 3. A statistically significant negative correlation was observed between MMO and 3D displacement (−0.41, *p* = 0.05), and rotation in the *Z*-axis (−0.560, *p* = 0.01).

## 4. Discussion

The quantitative 3D CT measurement method we have described here expands the classification by Loukota and Spiessl and Schroll quantitatively. This 3D method provides precise and detailed information about fractures of the condylar process, and the reproducibility is excellent.

In a previous study, the repeatability of the 2D measurement of ramus height loss measurement was reported to be excellent using OPT [18]. In our study, however, the 2D reproducibility was good only for ramus height loss measured on OPT. This discrepancy could be related to the difficulty in finding the highest point (reference point) of the condylar head in the fractured side on an OPT in case of severely displaced and dislocated fractures (Figure 2) due to rotational movements of the fracture fragments.

Measurement was impossible on 2D reconstructed CT and CBCT images, which was previously reported by Neff et al. [7]. Our study supports this finding. Results varied depending on the slices which were chosen for the measurements. Ramus height loss can be measured on the OPT with only fair reproducibility. Moreover, this provides limited information about fracture severity. In contrast to ramus height loss measurement based on OPT images, the 3D displacement measurements of the fracture of the condylar process are precise and hardly observer dependent.

To the best of our knowledge, this study provides the first objective description of reproducible measurement of displacement and rotation of any condylar fracture. We found the largest fracture 3D displacement and rotations in the condylar neck and diacapitular fractures according to the Loukota classification, and in the dislocated high condylar and diacapitular fractures without loss of ramus height according to the Spiessl and Schroll classification (type V and type VI A). A possible explanation could be the attachment of the lateral pterygoid muscle to the condylar neck and condylar head. Although the 3D measurements are precise, objective, and reproducible, it is not known at this point whether the large 3D displacement and rotation of the condylar neck and diacapitular fractures are clinically relevant or not. To answer this question a prospective study that takes into account additional clinical variables such as lateral excursion of mandible, pain, as well as mandibular function impairment, is needed.

Limitations of this retrospective study were missing data about MMO as well as Towne projections and OPT. Within these limitations, we found a low negative correlation between MMO and 3D displacement and moderate correlation with rotation on the *Z*-axis. A possible interpretation is that the 3D displacement or rotation on the *Z*-axis, regardless of fracture classification, might be inversely related to the mouth opening. As this mouth opening was measured when the CT scans were taken, i.e., shortly after trauma, the importance of this initial mouth opening parameter to the final functional outcome is unclear.

Compared to the Spiessl and Schrol classification of condylar fractures, the classification of Loukota et al., which was proposed as an aid for making treatment decisions [12], is simplified. For example, Loukota does not take displacement, dislocation, and deviation of the fractures into account, whereas these aspects are included in the Spiessl and Schroll classification. Another consideration is that many languages do not differentiate between displacement and dislocation, which makes it difficult to translate these terms. Therefore, these two concepts must be clearly defined: dislocation means that the condyle is outside the fossa; displacement indicates the separation between the fracture fragments [6]. Based on the 3D measurements, it is possible to quantify the dislocation, displacement, and deviation in the *X*, *Y*, and *Z*-axes.

With regard to the validation of the Loukota classification, disagreement between the observers occurred with two condylar fractures. The first observer judged these two fractures to be a base fracture, whereas the second observer judged them to be a neck fracture of the condyle. Regarding the classification of Spiessl and Schroll, the disagreement also occurred with the same patients. The disagreement involving the first fracture was due to a discrepancy in assessing low and high condylar fractures, which was a similar problem with the Loukota classification (neck and base). The disagreement on the second fracture regarding the classification results from a discrepancy in assessing type I and type II displacement of the condylar fractures. There is no clear objective cut-off between these two types. Similar disagreement on the assessment of the condylar fracture classification was reported previously regarding neck and base fractures [30]. Together with the classification systems of Loukota and Spiessl and Schroll, the results from the quantitative 3D measurements in the present study have contributed to the objective description of condylar fractures.

The volume measurement of the fragment(s) of condylar fractures has not yet been included in any classification. This measurement not only enables visualization of the fragmentation but also provides information about the extent and inter-fragmentary stability of the fracture. It also allows an estimation of the dimensions of the osteosynthesis materials to be chosen for fixation. Although these measurements can now be performed with the 3D method, it is unclear whether this would be helpful in the clinical setting.

## 5. Conclusions

The quantitative 3D measurements provide precise, objective, and reproducible information about the condylar fracture with regard to volume, dislocation, and rotation of fragments, with excellent reproducibility. The quantitative 3D measurements enable surgeons to classify the fractures more exactly in accordance with the classification of Loukota and Spiessl and Schroll. Further research should be conducted with 3D quantitative measurements to determine if this method could be used to support treatment choice and, more importantly, to predict the functional outcome of condylar fractures.

## Figures and Tables

**Figure 1 jpm-12-01225-f001:**
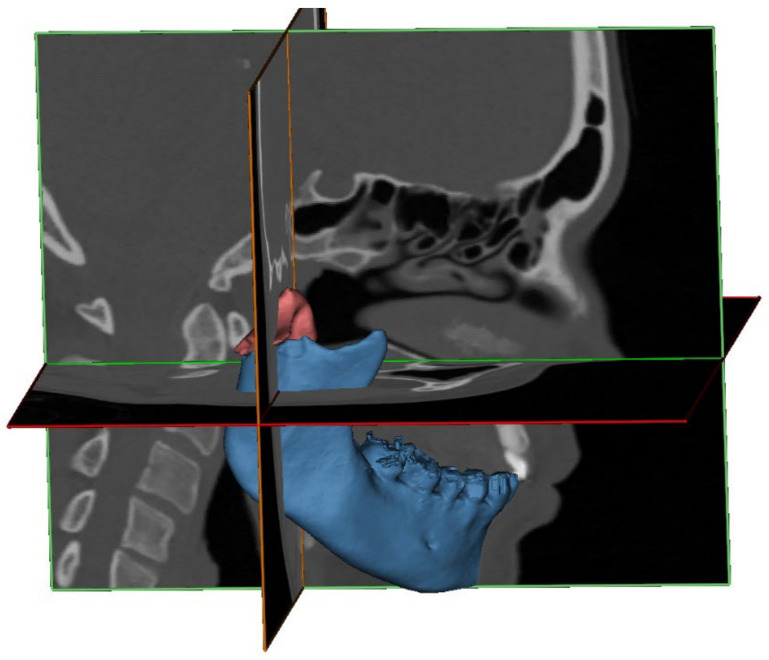
A 3D representation of a segmented fractured condyle (red) and mandible (blue), which is based on the CT scan, superimposed on the CT slices.

**Figure 2 jpm-12-01225-f002:**
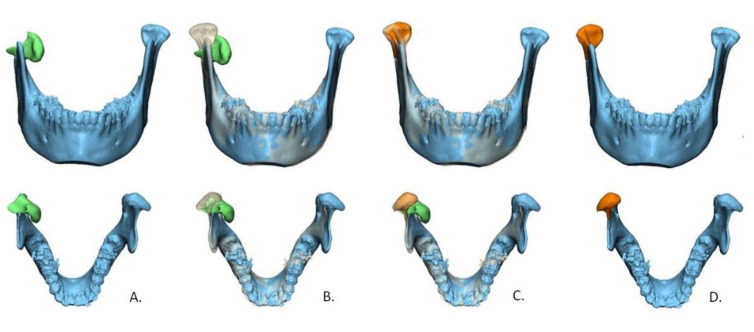
Workflow of virtual reduction in the fracture using a 3D model. (**A**) segmented condylar fracture fragment (green) from the mandible (blue); (**B**) template mirrored from non-fractured contralateral condyle; (**C**) fractured condyle is aligned according to the template; (**D**) virtually reduced model with the fracture fragment after reduction (orange).

**Figure 3 jpm-12-01225-f003:**
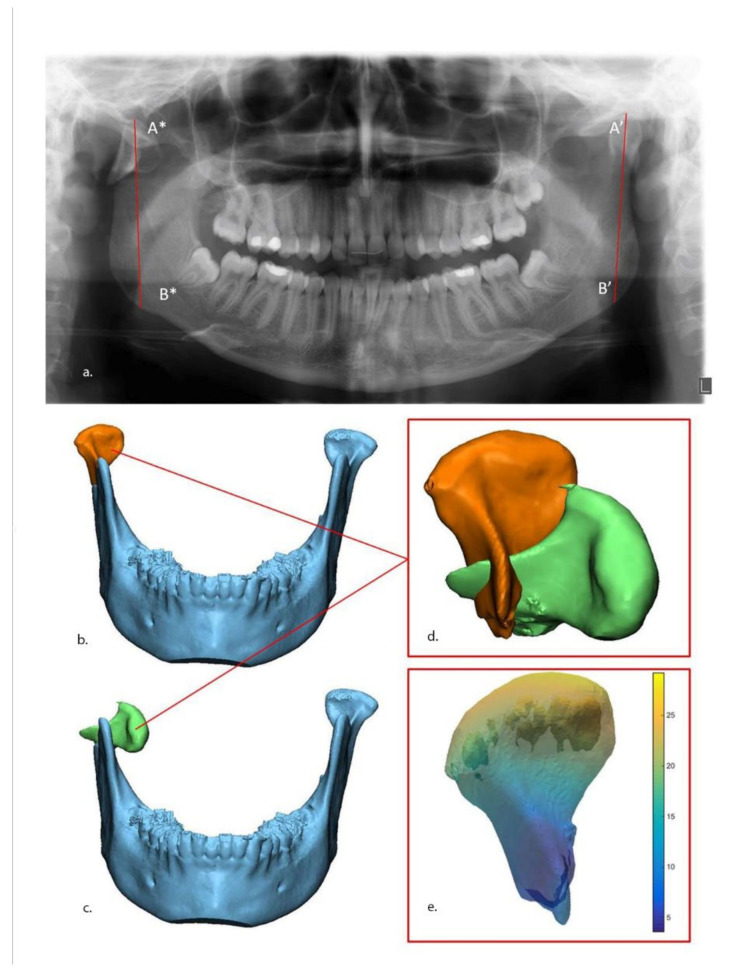
Measurements of ramus height loss and 3D displacement. (**a**) Ramus height loss measurement on panoramic: Loss of vertical height is the difference between non-fractured (A’–B’) side and fractured side (A*–B*). The 3D displacement of fractured condyle fragment after (orange) (**b**) and before (green) (**c**) virtual reduction. The two fragments (**d**) were imported into MATLAB software, after which a quantitative map of the displacement was calculated (**e**).

**Figure 4 jpm-12-01225-f004:**
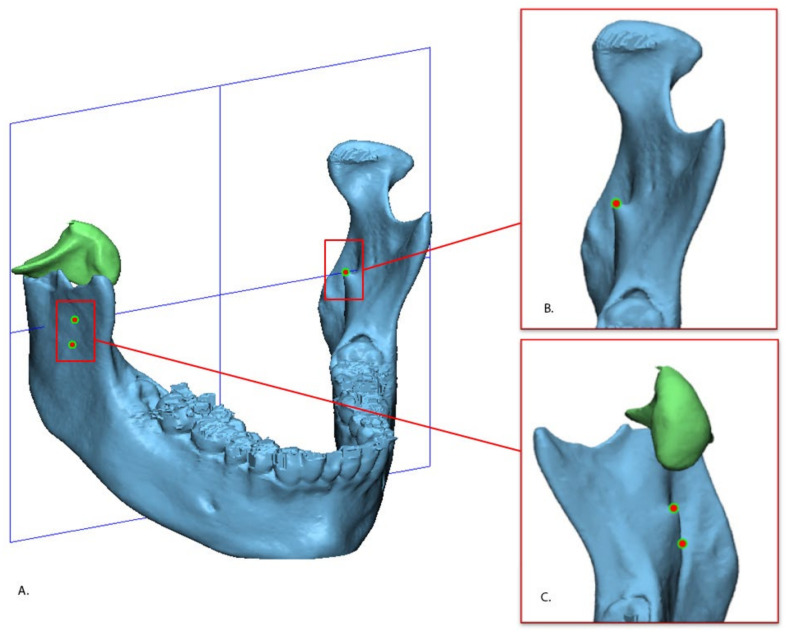
Defined standardized axis of the mandible (**A**). The axes was defined by using three points: The uppermost part of left lingual (**B**) and the uppermost and lowermost parts of the right lingula (**C**).

**Figure 5 jpm-12-01225-f005:**
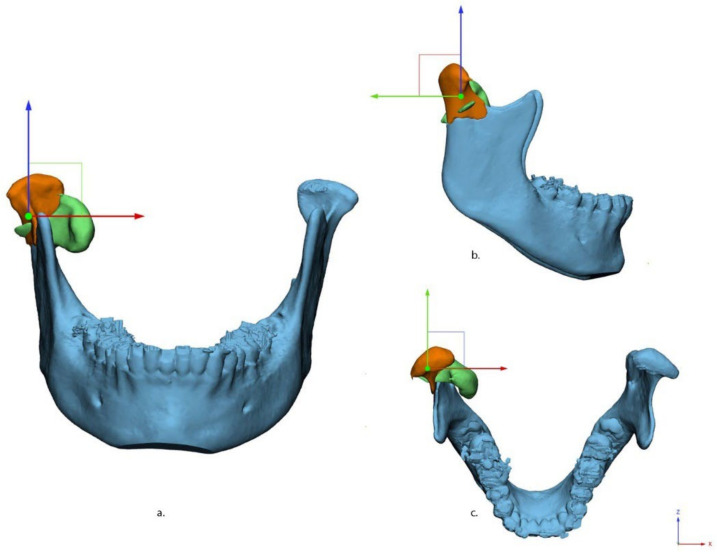
Fracture alignment in three views: (**a**) anterior–posterior, (**b**) medio–lateral, and (**c**) cranial–caudal. X (red), Y (green), and Z (blue) axes are shown in different colors.

**Table 1 jpm-12-01225-t001:** Measurements according to the Loukota classification.

Variables		Diacapitular Type A	Diacapitular Type B	Diacapitular Type C	Diacapitular Overall	Neck	Base	*p* Value *
MMO [mm; mean ± SD]		23.5 ± 10.9	26.5 ± 12.0	9.0	22.2 ± 10.9	16.5 ± 5.0	24.1 ± 4.9	0.235
Ramus height loss [mm; median (IQR)]		-	-	-	-	-	5.7 (1.6;8.2)	-
3D displacement [mm; median (IQR)]		8.3 (3.7;12.1)	1.3 (0.0;1.3)	10.3 (8.8;10.3)	6.1 (2.1;11.2)	10.7 (2.8;15.1)	5.0 (1.9;5.9)	0.117
Rotation [degrees; median (IQR)]	X	10.2 (2.2;15.8)	1.6 (0.0;1.6)	25.8 (19.9;25.8)	5.8 (1.1;18.1)	6.3 (1.9;35)	4.5 (2.1;4.5)	0.880
	Y	−9.8 (−10.1;−3.5)	0.9 (0.6;0.9)	−1.5 (−2.1;−1.5)	−1.5 (−9.8;0.7)	−3.6 (−17.6;18.3)	−0.1 (−1.5;7.9)	0.498
	Z	7.5 (4.8;10.8)	4.1 (2.6;4.1)	13.8 (6.8;13.8)	6.8 (4.4;10.5)	19.7 (0.4;64.6)	2.1 (0.4;9.0)	0.187
Volume [mm3; mean ± SD]		693 ± 440	851 ± 201	761 ± 72	761 ± 298	2140 ± 626	2226 ± 571	0.001

* Based on the Kruskal–Wallis test. Abbreviations: Max Mouth Opening (MMO), interquartile range (IQR), standard deviation (SD), and millimeters (mm).

**Table 2 jpm-12-01225-t002:** Measurements according to the Spiessl and Schroll classification.

Variables		Type I	Type II	Type IV	Type V	Type VI A	Type VI B	*p* Value *
MMO [mm; mean ± SD]		23.5 ± 6.7	26.3 ± 6.1	21.4 ± 7.1	12.0 ± 4.2	23.5 ± 10.9	18.0	0.354
Ramus height loss [mm; median (IQR)]								0.143
3D displacement [mm; median (IQR)]		2.3 (1.3;4.1)	5.4 (2.0;6.2)	6.8 (4.6;11.2)	11.9 (8.8;11.9)	8.3 (3.7;12.1)	3.5 (1.3;3.5)	0.013
Rotation [degrees; median (IQR)]	X	2.4 (0.7;4.5)	4.3 (1.7;8.4)	9.4 (5.3;13.0)	31.8 (19.9;31.8)	10.2 (2.2;15.8)	1.0 (0.0;1.0)	0.021
	Y	0.1 (−2.6;1.9)	6.3 (−6.7;11.9)	−13.0 (−16.4;−2.0)	0.0 (−2.1;0.0)	−9.8 (−10.1;−3.5)	3.9 (0.6;3.9)	0.031
	Z	1.4 (0.4;3.1)	5.8 (0.3;14.6)	8.6 (0.3−24.2)	20.8 (6.8;20.8)	7.5 (4.8;10.8)	4.8 (2.6;4.8)	0.173
Volume [mm^3^; mean ±SD]		2291 ± 826	2151 ± 392	1950 ± 575	1244 ± 838	693 ± 440	793 ± 247	0.002

* Based on the Kruskal–Wallis test. Abbreviations: Max Mouth Opening (MMO), interquartile range (IQR), standard deviation (SD), and millimeters (mm).

**Table 3 jpm-12-01225-t003:** Correlation between maximum mouth opening (MMO) and parameters.

Parameters		MMO	
		Spearman’s Correlation Coefficient	*p* Value
Ramus height loss		0.57	0.23
3D displacement		−0.41	0.05 *
Rotation	*X*-axis	−0.27	0.23
	*Y*-axis	−0.54	0.82
	*Z*-axis	−0.56	0.01 *
Volume		−0.12	0.59

* Statistically significant results (Spearman’s correlation). Abbreviations: Max Mouth Opening (MMO).

## Data Availability

The data presented in this study are available on request from the corresponding authors.

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
