# Peer review of "Quantitative Three-Dimensional Computed Tomography Measurements Provide a Precise Diagnosis of Fractures of the Mandibular Condylar Process"

_jpm, 2022, doi:10.3390/jpm12081225_

Round 1

Reviewer 1 Report

Dear Authors,

The article: 'Quantitative 3-dimensional computed tomography measurements provide a precise diagnosis of fractures of the mandibular condylar process' was to develop a standardized 3D quantitative measurement method for mandibular condyle fractures with high inter-user reliability, to correlate the obtained results with the Loukota and Spiessl & Schroll classifications and with clinical parameters as ramus height loss and maximum mouth opening.

Punctuation mistakes should be corrected.  English language and style are fine/minor spell check required

The introduction is well written. 

Materials and methods

figure 5 - add descriptions with figure (on the same page)

Include a bioethical committee note in your materials and methods.

The p value should be written in italic

Discussion is clearly presented.

Add table with abbeviations before references.

References should be prepared according MDPI guidelines.

The article is carefully prepared, the assumptions and aims are supported by results.

 Article can be accepted after minor revision.

Reviewer 2 Report

  I appreciate the opportunity to review the article entitled “Quantitative 3-dimensional computed tomography measurements provide a precise diagnosis of fractures of the mandibular condylar process”. This research proposed a new way of 3D quantitative measuring for fractured condylar fragments, which could give better information to clinicians. I put some points I have noticed below at this time.  Hope this helps.

1) The explanations of Figure2 C and D should be described in the legend.

2) The characters in Tables 1 and 2  should be aligned. And Tables(Tables 1  and 2) across the pages are hard to see.

3) In 3D measurements, which and how many points were being used for its measuring? Were the measuring points determined by the observers or automatically by software? That information should be provided in detail in the section “2-4-3 D-measurement”.

4) Some patients enrolled in this research had accompanying other mandibular fractures. Is there any possibility that those other fractures affect your measuring data, especially on ramus height loss?

5) Recently, an article on new ramus height measurements was published (J. Pers. Med. 2022, 12, 1181. https://doi.org/10.3390/jpm12071181). Why don’t you use the new method in this  and future research?
